# Multicomponent Behavioural Intervention during Pregnancy to Reduce Home Exposure to Second-Hand Smoke: A Pilot Randomised Controlled Trial in Bangladesh and India

**DOI:** 10.3390/ijerph21040490

**Published:** 2024-04-17

**Authors:** Veena A. Satyanarayana, Cath Jackson, Kamran Siddiqi, Mukesh Dherani, Steve Parrott, Jinshuo Li, Rumana Huque, Prabha S. Chandra, Atif Rahman

**Affiliations:** 1National Institute of Mental Health and Neuro Sciences (NIMHANS), Bangalore 560029, India; chandra@nimhans.ac.in; 2Valid Research Ltd., Wetherby LS22 7DN, UK; cath@validresearch.co.uk; 3Department of Health Sciences, University of York, York YO10 5DD, UK; kamran.siddiqi@york.ac.uk (K.S.); steve.parrott@york.ac.uk (S.P.); jinshuo.li@york.ac.uk (J.L.); 4Institute of Population Health, Department of Public Health, Policy and Systems, University of Liverpool, The Elms Medical Centre, Liverpool L8 3SS, UK; mukesh.dherani@nhs.net; 5ARK Foundation, Dhaka 1212, Bangladesh; rumanah14@yahoo.com; 6Institute of Population Health, Department of Primary Care and Mental Health, University of Liverpool, Liverpool L69 3GL, UK; atif.rahman@liverpool.ac.uk

**Keywords:** trial, perinatal, passive smoking, second-hand smoke, pregnancy, postpartum, intervention, tobacco, behaviour change

## Abstract

Background: Pregnant women exposed to second-hand smoke (SHS) are at increased risk of poor birth outcomes. We piloted multicomponent behavioural intervention and trial methods in Bangalore, India, and Comilla, Bangladesh. Methods: A pilot individual randomised controlled trial with economic and process evaluation components was conducted. Non-tobacco-using pregnant women exposed to SHS were recruited from clinics and randomly allocated to intervention or control (educational leaflet) arms. The process evaluation captured feedback on the trial methods and intervention components. The economic component piloted a service use questionnaire. The primary outcome was saliva cotinine 3 months post-intervention. Results: Most pregnant women and many husbands engaged with the intervention and rated the components highly, although the cotinine report elicited some anxiety. Forty-eight (Comilla) and fifty-four (Bangalore) women were recruited. The retention at 3 months was 100% (Comilla) and 78% (Bangalore). Primary outcome data were available for 98% (Comilla) and 77% (Bangalore). Conclusions: The multicomponent behavioural intervention was feasible to deliver and was acceptable to the interventionists, pregnant women, and husbands. With the intervention, it was possible to recruit, randomise, and retain pregnant women in Bangladesh and India. The cotinine data will inform sample size calculations for a future definitive trial.

## 1. Background

There is growing evidence that second-hand smoke (SHS) exposure during pregnancy is associated with poor pregnancy and infant outcomes [1]; SHS exposure during pregnancy has been associated with a slightly increased risk of stillbirth, preterm delivery, and congenital anomalies, although the results have been inconsistent [2,3,4]. Furthermore, infants born to women exposed to SHS during pregnancy are more likely to have a low birth weight (pooled OR = 1.2, 95% CI 1.1 to 1.3) compared with infants not exposed to SHS [5]. Across prospective and retrospective studies, the mean birth weight of infants born to SHS-exposed women is estimated to be about 33–40 g less [4]. One study based on a sensitive assay for cotinine showed a birth weight decrement of 27.2 g (95% CI 0.6 to 53.7) per unit change in log cotinine, which represented a decrement of about 100 g between the highest and lowest cotinine quintiles [6].

Globally, it is estimated that more than a third of all women are regularly exposed to SHS [7]. Most SHS exposure among reproductive-aged women in low- and middle-income countries (LMICs) occurs at home where women spend most of their time. Estimates of SHS exposure at home have ranged from 17.8% in Mexico to 72.3% in Vietnam [8]. An analysis of nationally representative data from 42 LMICs between 2003 and 2009 found that the prevalence of self-reported SHS exposure during pregnancy ranged from 9.3% in the Dominican Republic to 82.9% in Timor-Leste [9]. In surveys conducted in antenatal care settings in nine developing countries, 17.1% (Democratic Republic of the Congo) to 91.6% (Pakistan) of pregnant women reported that smoking was permitted in their homes, which translates to frequent exposure to SHS indoors [10]. Previous literature also shows that pregnant women in the Southeast Asian countries studied had the highest probability of SHS exposure. This high SHS exposure during pregnancy in Southeast Asia, therefore, poses a particular challenge for LMICs [1]. Several reasons are cited for this increased SHS exposure, including poor awareness of the harms associated with SHS, patriarchal family structures, and lack of women empowerment [9]. Due to the lack of empowerment, women may not feel comfortable challenging male smoking behaviour, even if they are aware of the potential harm [9].

Pregnancy provides a window of opportunity for the entire family to change harmful behaviours, especially when the focus is the health of the foetus. In 2013, the WHO developed the first-ever guidelines for the prevention and management of tobacco use and SHS exposure during pregnancy [11]. The guidelines recommend that healthcare providers ask all pregnant women about exposure to SHS as early as possible in pregnancy and at every antenatal care visit. They also recommend that healthcare providers give pregnant women, their partners, and other household members advice and information about the risks to pregnant women due to SHS exposure, and wherever possible, to provide cessation support to partners and other household members.

Despite its importance, good quality research on interventions to reduce home exposure to SHS among pregnant women is sparse, especially in LMICs [11]. This is demonstrated by our recent systematic review [12], which assessed the effectiveness of behaviour change interventions to reduce SHS exposure among pregnant women. Only six studies met the inclusion criteria, and none were in Southeast Asia. These studies concluded that behaviour change interventions led to a reduction in SHS exposure or husbands quitting smoking, improved the knowledge regarding the harm caused by SHS, and enabled the taking of positive actions towards smoke-free homes. However, the study quality was moderate to low. For example, there was an absence of control arms and short follow-ups; only two studies used objective exposure measures, and one study reported an objective measurement of the health outcomes.

In this study, we present the findings of a pilot RCT with embedded economic and process evaluation components. The aims were to examine the feasibility and acceptability of (a) our study design, measures, and methods, including the delivery of a multicomponent intervention to reduce home exposure to SHS among non-smoking pregnant women; (b) an intervention to reduce home exposure to SHS among non-smoking pregnant women from two LMICs, India and Bangladesh; and (c) an estimation of the standard deviation of the proposed primary outcome measure, to inform sample size calculation for a prospective definitive trial.

## 2. Methods

### 2.1. Study Design

This trial was retrospectively registered at the International Standard Randomized Controlled Trial Number (ISRCTN) registry (ID ISRCTN18132255) and was conducted as per the CONSORT guidelines for Pilot and Feasibility Trials. An individual 1:1 pilot randomised controlled trial with embedded economic and process evaluation components was conducted. Detailed formative work (systematic review [12], qualitative interviews with pregnant women and husbands [13]), and theory and evidence-based intervention development [14] preceded this pilot trial, in accordance with the UK Medical Research Council’s guidance for the development and evaluation of complex health interventions [15].

### 2.2. Setting and Participants

The trial was conducted in two sites: Comilla (Bangladesh) and Bangalore (India). Comilla is a typical peri-urban district, about 100 km southeast of the capital, Dhaka, with a population of 5.4 million. Four community clinics which had at least one hundred pregnant women registered during the study period were selected purposively for the trial [16].

The Bangalore study site was an antenatal clinic located in the South Zone of Urban Bangalore city, which comprises 32 urban slums in the Bruhat Bengaluru Mahanagara Palike (BBMP) municipal areas. The total population of these slums is around 149,743, with a ratio of 916 females per 1000 males [17].

These sites are both low-resource settings, where we could recruit women with low-income and low-literacy backgrounds, which are associated with a high prevalence of SHS exposure [1].

### 2.3. Recruitment and Randomisation

In both sites, women who were 20 weeks pregnant or less and attending the selected clinics were screened for eligibility. The women was eligible if she met the following inclusion criteria: over 18 years old, had reported SHS exposure due to the husband smoking at home, resident in the area, and NicAlert positive. Women using tobacco (smoking or chewing tobacco) were excluded from the study. If a woman reported exposure to SHS due to her husband’s smoking, she was given an information sheet about the study, and written informed consent was sought. The women then provided a saliva sample to test for nicotine levels using NicAlert.

Saliva specimens were assayed using NicAlert^®^ test strips (Craig Medical Distribution, Inc., Vista, CA, USA), in accordance with the manufacturer’s instructions [18]. The tests results included seven zones representing a range of cotinine concentrations from 0 (0–10 ng/mL) to 6 (>1000 ng/mL). The manufacturer’s cut-offs ≥1 (≥10 ng/mL) for saliva indicated tobacco exposure. All the NicAlert^®^ strips were read by a researcher who was blind to the participants’ SHS exposure status. Women who self-reported SHS exposure at home but were negative according to the NicAlert (<1 cut-off or <10 ng/mL) were excluded from the study. They were told about the effects of SHS on pregnancy and the foetus, advised to avoid close proximity with family members who smoked, and encouraged to consult staff in the antenatal clinic should they want more information or advice regarding this issue in the future.

Women who either self-reported or were observed to be showing signs of distress due to the intervention or study procedures were supported by the research staff. This typically involved providing accurate information on SHS, reassurance, and emotional support. Women who self-reported or were observed to be experiencing distress (anxiety, depressive symptoms, etc.) due to other causes (family, work stress, etc.) were referred to nearby mental health service providers.

For all the eligible women who provided written informed consent, baseline data (demographics, primary, and secondary outcomes) were collected in the community/antenatal clinic from October 2017 to January 2018. They were subsequently randomised to one of two arms: intervention or control. Randomisation was conducted using a computer-generated sequence sealed in opaque envelopes to conceal the arm allocation. The envelopes were prepared by a researcher who was not at the study site, which allowed equal numbers across the recruitment areas in each of the intervention arms. The participants’ details are presented in Figure 1 and Figure 2.

### 2.4. Intervention

Our theory- and research-based multicomponent behavioural intervention is described in detail elsewhere [14]. The components of the intervention were: (a) a pictorial intervention booklet with key messages on the effects of SHS on pregnancy and the foetus and how the risks could be minimised; (b) a cotinine feedback report about the presence of cotinine in the pregnant woman’s saliva sample; (c) a letter from the unborn baby to the father describing the harmful effects on him/her and on the mother due to the father’s smoking at home and requesting that he smoke outside; and (d) an automated and standardised voice calls to the mobile phone of the woman’s husband (only delivered in the Bangalore, India, site as it is a known information technology and communications hub and had the capacity and connectivity to support this mhealth component). Each woman was asked to seek permission from her husband to share his phone number and to receive a phone call from the research team. In this conversation, we sought the husband’s consent when asking him to receive the voice calls. These voice calls were delivered four times at fixed time intervals (one week after baseline, one week after the first call, two weeks after the second call, and one month after the third call). The voice messages were sent to the husband’s mobile phone to reinforce the key messages of the intervention. The total duration of the intervention was 9 weeks in the Bangalore site due to the additional component of voice calls, compared with a single-session intervention in the Comilla site.

The intervention was delivered at the community/antenatal clinic by two research staff (referred to as interventionists) in each site, a week after baseline assessments and randomisation. The interventionists were trained using a standard protocol to deliver the intervention and to ensure fidelity and consistency. An interventionist had one face-to-face session with the pregnant woman and provided the woman with a resource pack containing the multicomponent intervention and information about the contents of the pack and how it should be used. The women were encouraged to show the resource pack to their husbands and other family members and to discuss it with them. All the materials were developed and piloted for low-resource (low-literacy and low-income) settings. Simple language at the level of a primary school education was used, supplemented with relevant graphics to further facilitate comprehension for those with no or low literacy skills. Voice calls rather than text messages were used. The interventionist also read out each intervention component. The women were encouraged to seek similar help from family members to read the information at home.

### 2.5. Control

The women in the control arm received an educational leaflet that was developed for the study using standard facts about the ways that SHS harms the health of the pregnant woman and foetus. A limited attention control arm was considered ethically more appropriate than a placebo control arm for this vulnerable sample of pregnant women.

### 2.6. Primary and Secondary Outcomes

The first follow-up was three months after the intervention. The second follow-up was 24–48 h after delivery, when the birth outcomes were recorded. Saliva cotinine was the primary biological outcome indicating the level of SHS exposure and was collected at the baseline and at the three-month follow-up. To overcome the limitations of previous studies [12], which relied on self-reported outcomes following interventions to reduce SHS exposure, we used salivary cotinine, an objective biological measure, as a primary outcome. The research staff collecting data on the outcomes were blind to the allocation. The participants were given a food packet and compensated for travel or lost wages during the follow-up assessment.

The secondary outcomes were collected via a researcher, who administered a knowledge, attitude, and behaviour questionnaire at the baseline and at the three-month follow-up.

The questionnaire was based on existing scales in recent reviews [12,19,20]. The first section included questions on the women’s knowledge regarding SHS (e.g., Tobacco smoke inhaled by you is safe for your health—answer with yes, no, or don’t know). The second section included attitudes to SHS and their behaviours to reduce SHS exposure (e.g., I move away from my husband when he is smoking). The third section explored perceived confidence in negotiating change with husbands and other family members. And the last section was based on an integrative model of change [21] explored the perceived readiness of their husbands to change (e.g., Pre-contemplation: My husband is currently NOT considering changing his behaviour regarding smoking inside our home). The responses to these statements were collected using a 5-point Likert scale response system.

A third follow-up was carried out only in the Bangalore site to assess birth outcomes three months after delivery.

### 2.7. Health Economics

The health economics component was designed to pilot and evaluate a service use questionnaire in readiness for an economic evaluation within a fully powered RCT. The service use questionnaire (administered at three months) asked about the participant’s utilisation of health care services, including contacts with doctors and hospitals. Health-related quality of life was assessed using the EuroQol 5-dimension questionnaire and the version with 5 response levels (EQ-5D-5L) [22] in order to evaluate completion rates and enable the computation of quality-adjusted life years (QALYs).

### 2.8. Process Evaluation

The mixed-method process evaluation was conducted at the three-month follow-up. It comprised a brief survey for the pregnant women (24 intervention, 24 control Comilla; 21 intervention, 20 control Bangalore), interviews with the purposively selected pregnant women (10 Comilla, 10 Bangalore) and husbands (7 Comilla, 10 Bangalore) in the intervention arm, and four interventionists (2 Comilla, 2 Bangalore). The engagement with the intervention components and their acceptability and perceived impact were explored with the intervention arm participants. Feedback on the trial measures and methods was gathered from both the intervention and the control arm participants.

### 2.9. Statistical Analysis

The data were analysed using SPSS version 22. The saliva cotinine values and the questionnaire data were summarised using descriptive statistics. Independent sample *t*-tests and chi-square tests were used for the continuous and categorical data, respectively, to compare the two arms. The process evaluation survey data were summarised using descriptive statistics. The interviews were audio-recorded, transcribed verbatim, and translated into English for content analysis. The quantitative and qualitative datasets were triangulated using a matrix [23]. Excel and NVivo version 10 facilitated the data management. For the health economics-related analyses, we evaluated the completion rates and the utilisation of health care services by the trial participants. The data pertaining to quality of life were presented as frequency (%) of participants endorsing the extent of the problems across various domains of EQ-5d-5L: mobility, self-care, usual activity, pain/discomfort, and anxiety.

## 3. Results

The trial was carried out between October 2017 and January 2018; we recruited 48 women in Comilla (Bangladesh) and 54 women in Bangalore (India). Our retention rate at the three-month follow-up was 100% in Comilla and 78% in Bangalore (see Figure 1 and Figure 2).

### 3.1. Feasibility and Acceptability of Study Design, Measures, and Methods

It was possible to recruit (and retain) the women meeting the inclusion criteria during the time frame of the study. The pregnant women were willing to provide saliva for estimation of cotinine exposure as part of the screening for inclusion and for the outcomes three months after the intervention.

The women’s feedback indicated the acceptability (good or excellent) of the study information (96% Comilla, 88% Bangalore), the study questionnaires (81% Comilla, 91% Bangalore), and the meeting with the interventionist (88% Comilla, 91% Bangalore). The women in Comilla were less positive about providing the cotinine sample (50% rated this as fair or poor). This appeared to be linked to the anxiety-provoking test results rather than the procedure itself. Eighty-eight percent of the women in Bangalore rated this as good or excellent. Several of the women interviewed, and most of the husbands, spoke of the importance of hosting the study in the community/antenatal clinic as they saw this as a safe and credible setting.

### 3.2. Feasibility of Intervention Delivery

The interventionists were able to provide the face-to-face intervention, and the women agreed to take the intervention resource pack home.

Out of 98 voice calls made to the husbands, 62 (63.2%) calls were answered, and 50 (51%) calls were listened to completely (16–20 s). Of the 27 husbands to whom voice calls were made, 10 husbands listened to all 4 calls, 9 listened to 2 calls, and 8 listened only once. In summary, at least 50 calls were listened to completely, and all the husbands listened to at least 1 call.

### 3.3. Engagement with, and Acceptability of, the Intervention Components

Most of the pregnant women in the intervention arm engaged with the booklet, cotinine feedback report, and letter from the unborn baby (booklet: 80% Comilla, 91% Bangalore; report: 96% Comilla, 86% Bangalore; letter: 91% both sites) and discussed them with their husbands. Just half (48%) of the women in Bangalore listened to the voice messages sent to their husbands. All but one of the interviewed husbands in Comilla read the report, compared to only half in Bangalore. Conversely, half of those in Comilla saw the letter, whereas all the husbands in Bangalore saw it. The husbands often mentioned that their wives or other family members had read the information to them because they could not read. Most men seemed to have listened to one or two voice messages, though sometimes did not listen to the full message. Among the reasons offered for not listening were that they had recently changed their mobile number, had not received any calls, or were working/driving when the call came in.

The first three intervention components were highly rated (good or excellent) by those women who had read them (booklet: Comilla 88%; Bangalore 95%; report: Comilla 83%; Bangalore 70%; letter: Comilla 96%, Bangalore 94%). Most of the women interviewed found the booklet easy to understand. A small minority in Comilla said it was too long with too much text; this was confirmed by the interventionists in that site. The favourite parts of the content included the explanation of the risks of SHS; the picture of a happy family when the husband had stopped smoking at home; the text “oh, how good it feels to breathe in fresh air”; and the story-telling approach. Despite the positive rating, some women found the cotinine report to be difficult to understand; however, they all realised that their results indicated a problem with their health which they described as shocking, upsetting, frightening, and anxiety-provoking (Quotation 1, Table 1). An interventionist described the women as feeling “demoralized” and concerned that their unborn baby was already damaged by the SHS. They considered the report to be the “most challenging” part of the intervention to explain to the women, in terms of both technical and emotional content. In contrast, the women unanimously liked the letter, in particular the direct request from the unborn baby to the father (Quotation 2, Table 1). Its emotional nature was frequently mentioned, with the interventionists also describing the women’s emotional reaction. Finally, the voice calls were rated as good or excellent (82%) by the women who heard them; they had little to say about the content.

The husbands also commonly described the booklet as informative and the letter as emotional (Quotation 3, Table 1). Notably, those who went on to stop smoking in the home spoke confidently of understanding that the “red box” in the cotinine report indicated “harmful elements” in the bodies of their wives which lasted several hours and were caused by their smoking. This seemed to be new information to them (Quotation 4, Table 1). A few husbands remembered the voice calls to be about telling them to not smoke, to not smoke at home, and that smoking was “a problem for pregnant women” (IND 85 Successful husband).

### 3.4. Perceived Impact of the Intervention

In responding to the question “overall, how useful was the IMPRESS programme in achieving a smoke free home?” the women in Comilla reported a mean score of 4.96 (SD = 1.85, 1 = not at all useful, 7 = extremely useful). The women in Bangalore rated it more highly (mean = 6.05, SD = 1.43). Many of the interviewed women and husbands described how the intervention had prompted the husband to immediately state an intention to stop smoking in the home; some husbands stated that they would try to stop smoking completely. This motivation was clearly associated with protecting the health of the unborn baby (and, relatedly, the health of the pregnant woman) and was triggered differently by the intervention components. The booklet provided new knowledge about the risks of SHS, enabling some women to feel confident about approaching their husbands about their smoking, whereas the letter and cotinine report both prompted an emotional response, sorrow amongst the husbands and anxiety amongst the women, respectively (Quotations 1 and 5, Table 1).

The women whose husbands continued to smoke in the home offered several reasons for this continuation: the ignoring of the information in the study, addiction, forgetfulness (sometimes due to alcohol), wanting to relax after work, and the fear of being seen smoking outside by her parents (Quotations 6 and 7, Table 1). A small minority had not spoken to their husbands about smoking outside, as they saw this request as culturally inappropriate, feared their husband’s response, or did not believe it would make any difference (Quotation 8, Table 1). The interventionists suggested that some women “lack courage” to have these conversations for the same reasons. Finally, the husbands who did not change also mentioned addiction, smoking “unmindfully”, and finding it hard to go outside to smoke, when they tried, particularly in cold weather.

### 3.5. Baseline Characteristics

The combined baseline characteristics of the pregnant women and their husbands from both sites are shown in Table 2. The two sites were comparable with regard to most of the socio-demographic data. Just one significant difference was noted regarding alcohol use; in Bangalore, 22.8% (23) of the husbands were reported to be using alcohol, while none of the husbands in Comilla used alcohol.

### 3.6. Primary Outcome

Primary outcome data were available for 88 women (Bangalore: intervention (n = 26), control (n = 27); Comilla: intervention (n = 24), control (n = 24). In the Bangalore site, the saliva cotinine levels decreased significantly from the baseline to the three-month follow-up in both the intervention (baseline (N = 26) = 10.78 ± 5.22; follow-up (N = 21) = 3.1 ± 6.84; *p* < 0.001) and the control arms (baseline (N = 27) = 12.84 ± 7.03; follow-up (N = 20) = 5.97 ± 9.47; *p* < 0.01). In the Comilla site, the saliva cotinine levels decreased significantly from the baseline to the follow-up only in the control arm (baseline (N = 24) = 0.27 ± 0.23; follow-up (N = 24)= 0.14 ± 0.12; *p* = 0.002) and not in the intervention arm (baseline (N = 24) = 0.43 ± 0.42; follow-up (N = 23)= 0.59 ± 1.34; *p* = 0.144) (see Table 3).

### 3.7. Secondary Outcomes

In both arms, the pregnant women’s scores on knowledge about SHS and the behaviours related to their husbands’ smoking were similar (15.78 ± 4.01 vs. 15 ± 4.18) (21.60 ± 8.51 vs. 22.96 ± 10.28). The mean scores related to ‘readiness to change’ were also similar between the two arms 36.54 ± 10.73 vs. 34.49 ± 11.38 (Table 4).

In both sites, three months after intervention, more participants in the intervention arm reported that their husbands had stopped smoking at home (60% vs. 28.3%, OR: 8.31; 95% CI (1.993, 34.636) compared with those in the control arm. While there was no difference between the two arms in knowledge scores or attitudes related to smoking at home, more women in the intervention arm (65.1%) felt confident about negotiating behaviour change with their husbands (65.1% in the intervention arm compared with 33.3% in the control arm) [(OR: 3.29; 95% CI (1.388, 7.819)] (Table 4). More women in the intervention arm reported that their husbands were in the maintenance phase of change, while more husbands in the control arm were reported to be in the contemplation or pre-contemplation stages.

### 3.8. Birth Outcomes

The second follow-up at birth revealed that there was one early neonatal death (intervention arm) and that over two-thirds of the preterm births (intervention = 10 (71.4%) vs. control = 15 (88.2%)) were documented in Comilla. Bangalore documented all the live births and less than a third of the preterm births (intervention = 6 (26.1%) vs. control = 5 (21.7%)). A third follow-up 3 months after delivery, conducted only in Bangalore, revealed two infant deaths in the control arm. There were no reported adverse effects related to study participation in either Comilla or Bangalore.

### 3.9. Health Economics

The completion rates for the questionnaire were high, with all the headline questions being answered. Table 5 presents the responses to the main categories. In the control arm, 20% were visited by a doctor for pregnancy-related complications, compared with 24% in the intervention arm. Doctor visits for other reasons were made by 25% of the control and 26% of the intervention arm. Only one of the participants’ husbands received medication for breathing problems. None of the participants’ husbands sought smoking cessation help in the time period of the trial. Hospital visits for pregnancy complications were made by 11% of the control and 2% of the intervention arm. One overnight stay was recorded for pregnancy complications, and no overnight hospital stays were recorded for other reasons. All the participants completed an EQ-5D-5L.

Table 6 shows the frequencies for each of the five levels across all five domains. Notably, none of the responses included a level 5 response, which indicated the inability to perform any of the activities. Except for pain/discomfort, over half of the participants reported experiencing no problems in the other domains.

## 4. Discussion

The IMPRESS pilot study demonstrated the feasibility and acceptability of a multicomponent behavioural intervention to reduce home exposure to SHS in pregnant women in India and Bangladesh. The study design and procedures were found to be acceptable and indicated that a future definitive trial might be feasible. No significant intervention effects were found in the primary outcome (salivary cotinine levels); however, due to the nature of the trial, no definitive conclusions can be drawn at this stage. In the analysis focusing on secondary outcomes, it was observed that a significantly higher percentage of participants in the intervention arm reported that their husbands ceased smoking at home (60% vs. 28.3%, OR = 8.31; 95% CI = 1.993, 34.636).

We were able to recruit and retain the majority of the pregnant women for the duration of the study. The intervention was well tolerated (no adverse events), and the study measures, including biological sampling, were acceptable. The interventions were implemented effectively within the antenatal/community clinics, as recommended by the WHO [11], without any significant problems pertaining to training, supervision, or study oversight. This evidence for feasibility and acceptability concurs with previous trials on theory-based behaviour change interventions to reduce SHS exposure at home in pregnant women conducted in different countries, though little evidence for their scalability was found [12]. This indicates that a prospective definitive trial to evaluate the efficacy of the IMPRESS programme in a pragmatic setting would be feasible.

In this study, the participants rated the different components of the IMPRESS programme favourably. The first three intervention components were highly rated (good or excellent) by the majority of the trial participants. In the qualitative interviews conducted for the process evaluation, the intervention content was easily comprehensible in both settings. This was due to the extensive formative work conducted to develop this multicomponent intervention [12,13]. This included an in-depth qualitative study of smoking behaviours at home among South Asian men and the opinions of their pregnant wives regarding the acceptability and therapeutic effects of various strategies to quit smoking [13]. This multicomponent behavioural intervention also leveraged the knowledge of previous trials to reduce the practice of smoking in households in different countries [12].

In this pilot trial, we also assessed the acceptability of different forms of assessments, including sampling for salivary cortisol levels. Previous studies have indicated a need for more objective biochemically based measurements of SHS exposure to measure the intervention response [12,19,20]. Our analyses indicate that biological sampling is indeed feasible. In addition to testing smoking-related outcome measures, birth outcomes, a service utilisation-related questionnaire, and a quality of life scale (EQ-5D-5L) were also tested for their acceptability. These were found to be acceptable to the trial participants. In addition to evaluating the feasibility for different outcome measurements, we also quantitatively analysed the outcome data to generate statistics for sample size calculations for a future definitive trial.

Regarding the efficacy of the intervention, the pilot nature of the trial and the small sample size limits our ability to conclusively test the hypotheses, indicating the need for a future definitive trial. However, the inferential tests revealed several useful insights pertaining to outcome measurement, potentially informing our future trial. Three months after the intervention, more participants in the intervention arm, compared with the control arm, reported that their husbands had stopped smoking at home. While there was no significant difference in knowledge or attitudes about smoking at home between the two arms, the women in the intervention arm felt more confident about negotiating behaviour changes with their husbands.

The study primarily focused on SHS exposure, which was measured by the levels of saliva cotinine in the mothers three months post-intervention. In the Bangalore site, there was a notable decrease in saliva cotinine levels from the baseline to the follow-up period, in both the intervention and control arms, indicating a reduction in SHS exposure. However, in the Comilla site, this decrease was significant only in the control arm, with no substantial change observed in the intervention arm. Several factors could explain these differing outcomes between the two sites. Firstly, a ‘floor effect’ in the Comilla sample might have limited any further reduction in SHS exposure through the intervention. Secondly, the indoor SHS exposure levels in Comilla could be inherently low, as the prevention of smoking in public places is not strictly enforced, unlike in Bangalore, where indoor smoking might be more prevalent due to strict outdoor smoking regulations. Thirdly, the cotinine concentration in Comilla samples might have been altered during freeze-drying and transport [18].

Another interesting observation was the discordance between the NicAlert and the LCMS values at the baseline in Comilla; this was potentially due to cross-reactivity between the cotinine metabolites and NicAlert and changes in the cotinine concentration during sample processing [18]. This discrepancy challenges the concordance between the NicAlert and LCMS values. Additionally, both sites observed an unexpected decrease in saliva cotinine in the control arms, which could be related to behavioural changes during pregnancy, or it may have been influenced by the knowledge of the NicAlert test results.

### Limitations of the Study

Several limitations should be carefully considered when interpreting the findings of the current study. Firstly, as a pilot trial, the small sample size limited our ability to conclusively test hypotheses and to investigate the underlying mechanisms relating to our findings. While the study demonstrated the feasibility and acceptability of the multicomponent behavioural intervention, a larger trial is needed to establish the efficacy and fidelity of these interventions. Secondly, the sample was limited to pregnant women from low-income urban slums (Bangalore) and peri-urban areas (Comilla). As such, the results may not generalise to other areas of South Asia, which include numerous ethnic, regional, and national variations. Thirdly, due to the nature of the interventions tested in this trial, the study interventionists could not be blinded to the treatment conditions in the way that the research assistants and study coordinator were. Fourthly, it is possible that the samples from Comilla may have been altered during freeze-drying and transport to the UK. Furthermore, future research should address contamination between the arms and the potential confounding variables that this randomised trial could not address or examine, such as exposure to environmental hazards, motivation for treatment, co-occurring alcohol use by the husbands, and parity.

## 5. Conclusions

We demonstrated the feasibility and acceptability of delivering and evaluating a brief, culturally tailored, gender intentional, theory- and research-based, multicomponent behavioural intervention to reduce home exposure to SHS exposure in pregnant women in Bangalore, India, and Comilla, Bangladesh. The intervention is scalable, requires limited training, and can be delivered by health workers in community and antenatal clinics. This RCT represents the first investigation in the South Asian region to report a detailed description of a multicomponent intervention and to use biological outcomes. However, a definitive trial in future will be able to provide more convincing evidence.

## Figures and Tables

**Figure 1 ijerph-21-00490-f001:**
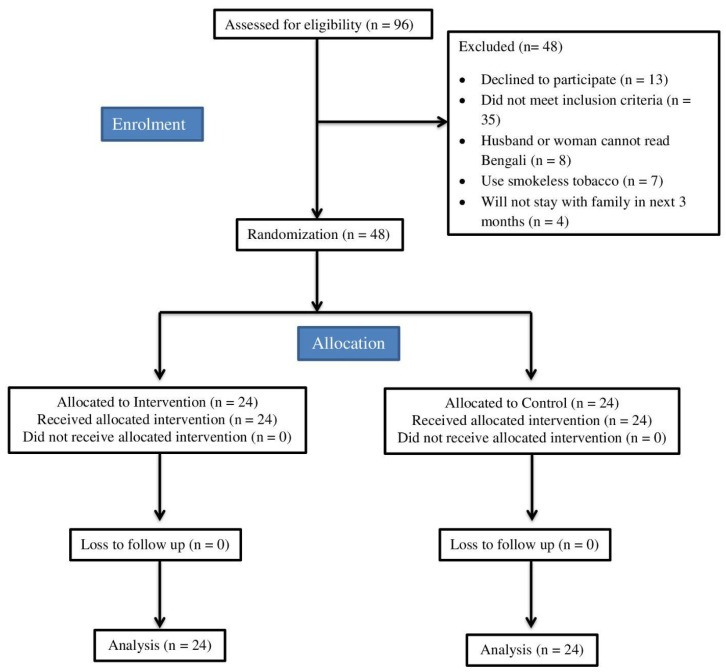
IMPRESS trial flow diagram—Comilla (adapted from consort 2010).

**Figure 2 ijerph-21-00490-f002:**
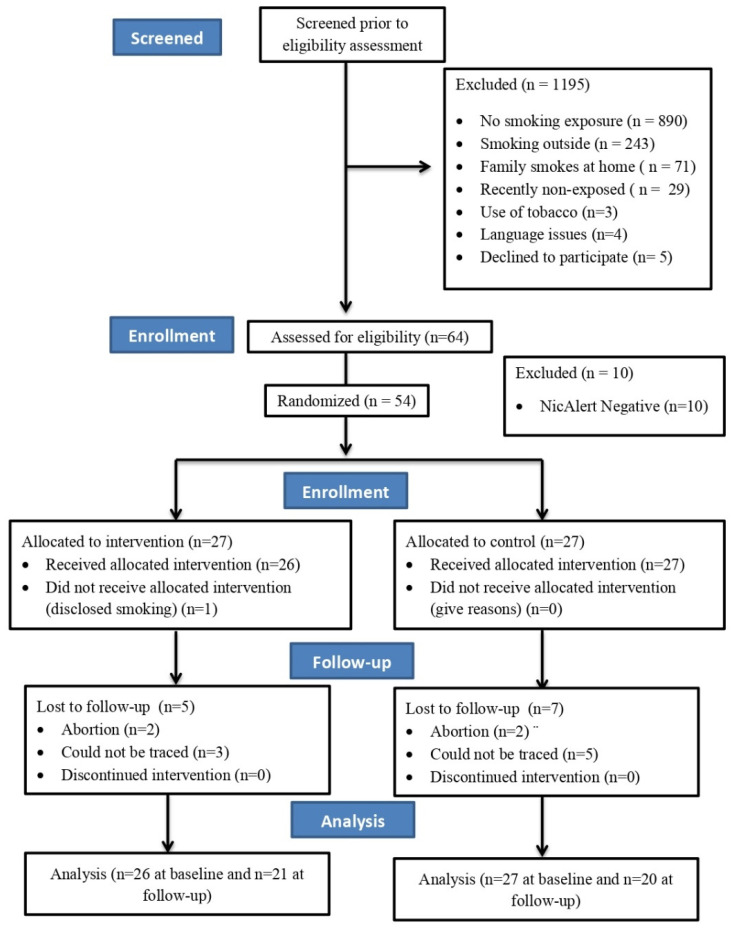
IMPRESS trial flow diagram—Bangalore (adapted from consort 2010).

**Table 1 ijerph-21-00490-t001:** Illustrative quotations.

ID	Quotations
1	*I was nervous when I got the report. My daughter said that ‘the baby can even die because of the smoke’, as was written in the report. I was anxious about my baby. When I told my husband about the report, he got nervous too. We are poor people, if anything happens, we cannot afford to go to big hospital. In this community clinic, we get medicine for fever, cough and cold only. The hospital is far from here.* Pregnant woman ID161, Bangladesh, husband now smokes outside
2	*I liked the way the baby was talking to my husband. He said (after reading it) I cannot stop completely but I will smoke outside home.* Pregnant woman ID33, India, husband now smokes outside
3	*The letter is very touching, it made me emotional. It is very nicely written.* Husband ID176, Bangladesh, now smokes outside
4	*The report indicated the presence of harmful ingredients in my wife’s body which came from the smoke of my cigarette. To be honest, I had no idea that harmful chemicals can stay in the body of my wife for my habit. I mean if I smoke in front of her, even after few hours. I knew that smoking harms the smoker. After seeing the report and when my wife said that a lady tested her saliva using a using a strip, I realised for the first time that the smoke coming out of a cigarette is very harmful.* Husband ID178, Bangladesh, now smokes outside
5	*After reading the letter, he was quiet for some time, he was thinking. Then he told me that “This is true, I should not smoke in front of you.” A smoker never likes anything against smoking, but the letter talked about the wellbeing of the child. I think that’s why he read it, was sorry and agreed that it was true.* Pregnant woman ID136, Bangladesh, husband now smokes outside
6	*He said (after reading the report), “doctors will say all these things, and it’s not true” If you talk to husband directly that’s good, otherwise if I say anything about this he will not listen*. Pregnant woman ID32, India, husband continues to smoke inside
7	*Usually he smokes outside that’s near the door only but sometimes he forgets and smokes inside the home also, then I have to keep reminding him about it. Even my first son who is three years old, tells his father “Go out and smoke, don’t smoke inside”.* Pregnant woman ID23, India, husband continues to smoke inside
8	*A husband should listen to his wife. A wife sacrifices so much for husband, leaves her family, cooks for him. But all husbands are not the same. Some do not listen to their wives. She should talk to her husband when he is in good mood (laughing). But this is not possible for everyone.* Pregnant woman ID131, Bangladesh, husband now smokes outside

**Table 2 ijerph-21-00490-t002:** Socio-demographic details of the pregnant women and husbands at baseline.

	Total N = 101	Intervention N = 50	Control N = 51	Sig.
Age	Mean ± SD	24.78 ± 5.258	23.92 ± 4.58	0.384
Education of Pregnant Women	Primary Schooling	20 (40%)	22(43.1%)	0.749
	Secondary Schooling and Above	30(60%)	29(56.9%)	
Education of Husband	Primary Schooling	23 (46%)	28 (54.9%)	0.243
	Secondary Schooling and Above	27 (54%)	23 (45.1%)	
Occupation of Pregnant Women	Employed	6 (12%)	12 (23.5%)	0.130
	Homemakers	44 (88%)	39 (76.5%)	
Occupation of Husband	Professional/Business	15 (30%)	12 (23.5%)	0.604
	Skilled/Unskilled Labours	33 (66%)	38 (74.5%)	
	Unemployed	2 (4%)	1 (2%)	
Type of Family	Nuclear	30 (60%)	27 (52.9%)	0.304
	Joint	20 (40%)	24 (47.1%)	
Gravida	Primi-	13 (26%)	15 (29.4%)	0.702
	Multi-	37 (74%)	36 (70.6%)	
Type of Fuel	Wood/kerosene	26 (52%)	30 (58.8%)	0.312
	LPG	24 (48%)	21 (41.2%)	

**Table 3 ijerph-21-00490-t003:** Saliva cotinine levels in pregnant women at baseline and follow-up.

Saliva Cotinine			Intervention (ng/mL)	Control (ng/mL)	Sig
Bangalore	Baseline	Mean (SD)	10.78 ± 5.22	12.84 ± 7.03	0.119
		N	26	27	
	Follow-Up	Mean (SD)	3.1 ± 6.84	5.97 ± 9.47	0.049
		N	21	20	
Comilla	Baseline	Mean (SD)	0.43 ± 0.42	0.27 ± 0.23	0.155
		N	24	24	
	Follow-Up	Mean (SD)	0.59 ± 1.34	0.14 ± 0.12	0.068
		N	23	24	

**Table 4 ijerph-21-00490-t004:** Details of smoking behaviour and SHS exposure at baseline.

Baseline Exposure to SHS at Home	Frequency	Intervention 50	Control 51	Sig.
Pregnant women’s reports on husbands’ smoking behaviour at home	Daily	44 (88%)	44 (86.3%)	0.477
	4–6 days in a week	3 (6%)	1 (2%)	
	1–3 days in week	3 (6%)	6 (11.76%)	

**Table 5 ijerph-21-00490-t005:** Indications of service use frequencies for the two study arms.

		Arm	
		Intervention	Control
		Count	Count
Pregnant woman visited at home by doctor (complications)	No	34	37
	Yes	11	9
Visited doctor for other reason	No	32	34
	Yes	13	12
Husband had medication for breathing problem	No	43	43
	Yes	1	3
Smoking cessation helped husband	No	45	46
	Yes	0	0
Pregnant woman, hospital no-stay (complications)	No	40	45
	Yes	5	1
Pregnant woman, hospital no-stay (no complications)	No	42	45
	Yes	3	1
Pregnant woman, hospital overnight stay (complications)	No	44	46
	Yes	1	0
Pregnant woman, hospital overnight stay (no complications)	No	45	46
	Yes	0	0

**Table 6 ijerph-21-00490-t006:** EQ-5D-5L by domain.

	Mobility		Self-Care		Usual Activity		Pain/ Discomfort		Anxiety	
	N	%	N	%	N	%	N	%	N	%
No problem	58	63.74	60	65.93	54	59.34	36	39.56	49	53.85
Slight problem	22	24.18	21	23.08	28	30.77	35	38.46	26	28.57
Moderate problem	9	9.89	8	8.79	9	9.89	18	19.78	13	14.29
Severe problem	2	2.20	2	2.20	0	0.00	2	2.20	3	3.30
Inability	0	0	0	0	0	0	0	0	0	0
Total	48	100.0	48	100.0	48	100.0	48	100.0	48	100.0

## Data Availability

The datasets used and/or analysed during the current study are available from the corresponding author on reasonable request.

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
