# Peer review of "Multicomponent Behavioural Intervention during Pregnancy to Reduce Home Exposure to Second-Hand Smoke: A Pilot Randomised Controlled Trial in Bangladesh and India"

_ijerph, 2024, doi:10.3390/ijerph21040490_

Round 1

Reviewer 1 Report

Comments and Suggestions for Authors

The study addressed one of the emerging and top public health agendas.

- The study gap is not clearly addressed other than the generic statement "Despite its importance, the research on interventions to reduce home exposure to 95 SHS among pregnant women is sparse, especially in LMICs". We expect some efforts would be available and summarizing them including their limitations and the body of knowledge this study could add might be useful.

- The reason for including semi-urban and slum areas needs to be described. Is the study focused on these areas and not aimed to provide a conclusion or generalization for the general population? If this is the case, this needs to be stated.

- The proper implementation of intervention could be affected by women's knowledge and understanding capability of the circulated information. Specifically, it is important to note and clearly address the educational level of the women as well as their husbands to do such intervention, e.g., sending letters from the unborn baby. I found the educational level in the results section, and about half had no education/primary. Thus, we need to know how the intervention in these groups was carried out, and the approach should be addressed in the methods section.

- Is informed consent taken from the husband to contact him via phone or send interventions?

- If the women feel stressed, did the ethics address this? The ethics principle lacks clarity and is better addressed in a separate section, e.g., women who tested but were not included in the study were only given some information. What actions were taken if the women stressed through the process?

- The interventions were not consistent across areas (e.g., including only phone intervention in India), and this could affect the outcome of the study. The reasons for using various interventions across different areas are not stated.

Author Response

Reviewer 1:

The study addressed one of the emerging and top public health agendas.

  1. The study gap is not clearly addressed other than the generic statement "Despite its importance, the research on interventions to reduce home exposure to 95 SHS among pregnant women is sparse, especially in LMICs". We expect some efforts would be available and summarizing them including their limitations and the body of knowledge this study could add might be useful.

We have added some text (lines 74-83) to be clearer on the limitations of the existing research reported in our systematic review e.g. no studies from Southeast Asia, lack of control arms, short follow-ups.

  1. The reason for including semi-urban and slum areas needs to be described. Is the study focused on these areas and not aimed to provide a conclusion or generalization for the general population? If this is the case, this needs to be stated.

We have added a sentence (lines 112-114) to explain that these are low resource settings, where we could recruit women with low income and low literacy backgrounds, which are associated with high prevalence of SHS exposure [1]. By focusing on these women, we can try to address health inequalities. We comment on the implications of this for generalisability in the discussion (line 465-467).

  1. The proper implementation of intervention could be affected by women's knowledge and understanding capability of the circulated information. Specifically, it is important to note and clearly address the educational level of the women as well as their husbands to do such intervention, e.g., sending letters from the unborn baby. I found the educational level in the results section, and about half had no education/primary. Thus, we need to know how the intervention in these groups was carried out, and the approach should be addressed in the methods section.

We have now clarified that our sample from both the sites are likely to be from low resource settings (low income and low literacy backgrounds) with high prevalence of SHS exposure (lines 112-114). We have also added text in the methods (line 174-180) providing detail on the careful development and delivery of all the intervention components (primary school literacy level, graphics, interventionists reading the information, voice messages not text messages). Formative qualitative work (Jackson et al, 2016) informed intervention development and the process evaluation indicated acceptability of these components by women. All the women and their husbands had a minimum of primary level school education (see updated Table 2).

  1. Is informed consent taken from the husband to contact him via phone or send interventions?

The point of first contact was the pregnant woman who provided ,written informed consent at recruitment (line 140-142). Women in India were requested to seek permission from their husband to share his phone number and receive a phone call from the research team. In this conversation we sought the husband’s consent for receiving the voice calls. Text has been added (lines 158-160).

  1. If the women feel stressed, did the ethics address this? The ethics principle lacks clarity and is better addressed in a separate section, e.g., women who tested but were not included in the study were only given some information. What actions were taken if the women stressed through the process?

We have added text on this (lines 129-139).

Women who self-reported SHS exposure at home but were negative on the NicAlert (<1 cut-off or < 10ng/ml) were excluded from the study. They were told about the effects of SHS on pregnancy and the foetus, advised to avoid close proximity with family members who smoked and. encouraged to consult staff in the antenatal clinic should they want more information or advice regarding this in the future.

Women who took part in the study who either self-reported or were observed to be showing signs of distress due to the study procedures or intervention were supported by the research staff. This typically involved, providing accurate information on SHS, reassurance and emotional support. Women who self reported or were observed to have distress (anxiety, depressive symptoms etc) due to other reasons (family, work stress etc) were referred to nearby mental health service providers (lines 129-139).

  1. The interventions were not consistent across areas (e.g., including only phone intervention in India), and this could affect the outcome of the study. The reasons for using various interventions across different areas are not stated.

Yes, only the Bangalore (India) site received an additional component of the intervention, that is, the voice calls made to the husbands. Unlike Comilla which was a peri-urban site, Bangalore was an urban site with good mobile phone connectivity. Further, Bangalore, is known to be an Information Technology and Communications Hub which made it feasible for us to test this component of the intervention (text added to lines 156-158). While this would affect the outcomes, our primary objective in this pilot trial was to test which components of the intervention are feasible and acceptable and not the effect of these on the outcomes.

Reviewer 2 Report

Comments and Suggestions for Authors

The paper is very well-written. I have some minor suggestions to improve its quality.

1. Page 2 of 15

Page number : 83

"women empowerment women"

There appears to be an extra "women" at the end of "women empowerment women.

2. Page 3 of 15

Line number: 143

"Women who were negative on the NicAlert were advised to be careful about the effects of SHS on pregnancy and the foetus and excluded from the study. "

Authors may need to briefly explain what the term "negative" means. Does it indicate that no cotinine concentration was detected?

3. Page 4 of 15

Line number: 180-182

"To overcome the limitations of previous studies, salivary cotinine, a biological measure, was used as a primary outcome."

What are those limitations? Authors may need to elucidate these limitations from the previous studies.

4.  Page 4 of 15 and Page 10 of 15

Page 4 of 15, Line number: 173-175

"Control :  Women in the control arm received an educational leaflet developed for the study  using standard facts about harms from SHS on the health of the pregnant woman, and fetus. "

Is there any reason why the control group received an educational leaflet instead of a blank leaflet containing no information about harms from SHS (a placebo group)? As shown in Table 3 (Page 10 of 15), the intervention effects were non-significant, and it might be due to the control group also having received certain education.

Is it because of ethical issues that the control group must also have received an education leaflet? Authors can briefly justify the assignment of the control group.

5. Page 13 of 15

Line number: 431-435

"Secondly, .....and transport"

This paragraph is very well-written. It effectively addresses the confusion regarding the differences in salivary cotinine levels between Bangalore and Comilla, particularly noting that Comilla exhibited much lower cotinine concentrations at baseline and follow-up compared to Bangalore.

It would be beneficial for the authors to include this explanation (Comilla samples might have been altered during freeze-drying and transport) as a potential limitation in the limitation section of the paper (page 14 of 15, Limitations of the study).

Author Response

Reviewer 2:

The paper is very well-written. I have some minor suggestions to improve its quality.

  1. Page 2 of 15

Page number : 83

"women empowerment women"

There appears to be an extra "women" at the end of "women empowerment women.

We have now deleted the extra word “women” at the end of the phrase (line 62).

  1. Page 3 of 15

Line number: 143

"Women who were negative on the NicAlert were advised to be careful about the effects of SHS on pregnancy and the foetus and excluded from the study. "

Authors may need to briefly explain what the term "negative" means. Does it indicate that no cotinine concentration was detected?

We describe in lines 126-127 “The manufacturer’s cut-offs ≥1 (≥10 ng/ml) for saliva indicated tobacco exposure.” “Negative” refers to values below this cut off irrespective of women’s self-reports of SHS exposure. We have added detail (line 129-133) to be clearer on this.

  1. Page 4 of 15

Line number: 180-182

"To overcome the limitations of previous studies, salivary cotinine, a biological measure, was used as a primary outcome."

What are those limitations? Authors may need to elucidate these limitations from the previous studies.

We have now clarified that an important limitation identified in our systematic review of behaviour change interventions to reduce SHS exposure in pregnant women was that very few studies used an objective health outcome, instead relying on self-report (lines 190-192).

  1. Page 4 of 15 and Page 10 of 15

Page 4 of 15, Line number: 173-175

"Control Women in the control arm received an educational leaflet developed for the study using standard facts about harms from SHS on the health of the pregnant woman, and fetus. "

Is there any reason why the control group received an educational leaflet instead of a blank leaflet containing no information about harms from SHS (a placebo group)? As shown in Table 3 (Page 10 of 15), the intervention effects were non-significant, and it might be due to the control group also having received certain education.

Is it because of ethical issues that the control group must also have received an education leaflet? Authors can briefly justify the assignment of the control group.

It is indeed for ethical reasons that a limited attention control group was considered more appropriate (rather than a placebo control group) in this vulnerable population of pregnant women. An educational leaflet containing facts about harms from SHS served this purpose. Text has been added on this (lines 184-185).

  1. Page 13 of 15

Line number: 431-435

"Secondly, .....and transport"

This paragraph is very well-written. It effectively addresses the confusion regarding the differences in salivary cotinine levels between Bangalore and Comilla, particularly noting that Comilla exhibited much lower cotinine concentrations at baseline and follow-up compared to Bangalore.

It would be beneficial for the authors to include this explanation (Comilla samples might have been altered during freeze-drying and transport) as a potential limitation in the limitation section of the paper (page 14 of 15, Limitations of the study).

We have now incorporated this important suggestion in the limitations section (lines 469-470).

Reviewer 3 Report

Comments and Suggestions for Authors

I reviewed the manuscript and found it to be intriguing, with a well-articulated focus. Nonetheless, there are minor issues necessitating the authors' attention. The pilot study endeavors to ascertain the feasibility of the intervention within a real Randomized Controlled Trial (RCT) framework.

The study's title lacks specificity, particularly concerning the intervention, which appears to be a multifaceted behavioral intervention targeting pregnant mothers to mitigate Secondhand Smoke (SHS) exposure.

The abstract appears overly verbose and requires truncation to align with the journal's guidelines.

There are typographical errors on line 127 and line 192 in the method section. Additionally, the full name of acronyms utilized at line 113 and 202 should be provided for clarity.

Considering the study criteria, how were confounding variables managed for women exposed to environmental hazards in workplaces or other confined spaces? What measures were implemented to prevent data contamination between the intervention and control groups?

The study implemented a multicomponent behavioral intervention strategy. Could you elaborate on the development and validation processes of the intervention? Furthermore, what is the duration of one complete cycle of the intervention, delivered over four cycles, and how was consistency in delivery maintained across interventionists?

Based on the information gleaned from Figures 1 and 2, alongside textual descriptions, there appears to be evidence of disparate implementation of the intervention, notably with the India team receiving additional intervention compared to the Bangladesh team.

The findings regarding health economics seem perplexing, particularly in Table 5, which solely delineates the healthcare service utility by participants. Conversely, Table 6 suggests evidence provided by participants regarding severe issues across EQ-5D-5L domains. How do you intend to reconcile this disparity?

Author Response

Reviewer 3:

I reviewed the manuscript and found it to be intriguing, with a well-articulated focus. Nonetheless, there are minor issues necessitating the authors' attention. The pilot study endeavours to ascertain the feasibility of the intervention within a real Randomized Controlled Trial (RCT) framework.

  1. The study's title lacks specificity, particularly concerning the intervention, which appears to be a multifaceted behavioural intervention targeting pregnant mothers to mitigate Second hand Smoke (SHS) exposure.

We have now specified the intervention as “Multicomponent Behavioural Intervention” and included “Home” exposure to SHS in the modified title (lines 2-3).

  1. The abstract appears overly verbose and requires truncation to align with the journal's guidelines.

We have made substantial edits to the abstract and it now meets (<200 words) the journal’s stated word limit.

  1. There are typographical errors on line 127 and line 192 in the method section. Additionally, the full name of acronyms utilized at line 113 and 202 should be provided for clarity.

Typographical errors have been corrected (lines 57, 110) and full name of acronyms are mentioned (lines 94-95 and 213).

  1. Considering the study criteria, how were confounding variables managed for women exposed to environmental hazards in workplaces or other confined spaces? What measures were implemented to prevent data contamination between the intervention and control groups?

Exposure to environmental hazards in workplaces or other confined spaces are important confounding variables. Although they were not systematically assessed in the present study, we assume their equal distribution in both groups in this RCT design. These are documented as limitations of our study (lines 472-473). Regarding contamination, since it was only a single session intervention at the clinic, chances of contamination was minimal. Beyond requesting women to not discuss the content of the intervention leaflet/booklet with other women in the clinic and their neighbourhood and encouraging them to discuss within their family, we were unable to do more (line 470-472).

  1. The study implemented a multicomponent behavioural intervention strategy. Could you elaborate on the development and validation processes of the intervention? Furthermore, what is the duration of one complete cycle of the intervention, delivered over four cycles, and how was consistency in delivery maintained across interventionists?

We are unable to include these details as they have been published elsewhere (lines 149-150). We have added to the text (lines 98-102) that we undertook a systematic review, qualitative interviews with pregnant women and husbands, and a theory and evidence-based approach to intervention development.

We have also clarified the duration of the intervention in both sites (lines 164-166) and added text (lines 169-170) about the training that interventionists received to ensure consistency in delivery.

  1. Based on the information gleaned from Figures 1 and 2, alongside textual descriptions, there appears to be evidence of disparate implementation of the intervention, notably with the India team receiving additional intervention compared to the Bangladesh team.

Yes, the India site received an additional component of the intervention, that is, voice calls made to husbands. Unlike Comilla which was a rural site, Bangalore was an urban site with good mobile phone connectivity. Further, Bangalore, is known to be an Information Technology and Communications Hub which made it possible for us to test this component of the intervention in our feasibility trial (lines 156-158).

  1. The findings regarding health economics seem perplexing, particularly in Table 5, which solely delineates the healthcare service utility by participants. Conversely, Table 6 suggests evidence provided by participants regarding severe issues across EQ-5D-5L domains. How do you intend to reconcile this disparity?

We don’t see these to be contradictory findings. EQ-5D-5L measures self-perceived health on the day of administering than rather clinically deemed needs. Therefore, scoring poorly on EQ-5D-5L does not warrant a clinical diagnosis which might result in use of health care services. Secondly, even if participants felt poorly and needed health care, they would be restricted by their individual circumstances, such as distance to care or financial burden. Their needs might not be transformed into actual use. Thirdly, the health care service use questions had the trial period as the recall period, which was not corresponding to the day when EQ-5D-5L was administered. It could be that participants felt unwell on the day of completing questionnaire but had no need to see a doctor before that during the recall period, or vice versa. When using multiple datapoints, the changes in EQ-5D-5L might be loosely associated with health care services use, but they are not closely associated. In the current study, it was only to test out the validity of the health care services scope and the feasibility of administering EQ-5D-5L. We have made a few edits for better clarity (lines 383-387).